# Exploring Entropy Measurements to Identify Multi-Occupancy in Activities of Daily Living

**DOI:** 10.3390/e21040416

**Published:** 2019-04-19

**Authors:** Aadel Howedi, Ahmad Lotfi, Amir Pourabdollah

**Affiliations:** School of Science and Technology, Nottingham Trent University, Clifton Lane, Nottingham NG11 8NS, UK

**Keywords:** activity recognition, independent living, activities of daily living, multi-occupancy, approximate entropy, sample entropy, fuzzy entropy, abnormality detection

## Abstract

Human Activity Recognition (HAR) is the process of automatically detecting human actions from the data collected from different types of sensors. Research related to HAR has devoted particular attention to monitoring and recognizing the human activities of a single occupant in a home environment, in which it is assumed that only one person is present at any given time. Recognition of the activities is then used to identify any abnormalities within the routine activities of daily living. Despite the assumption in the published literature, living environments are commonly occupied by more than one person and/or accompanied by pet animals. In this paper, a novel method based on different entropy measures, including Approximate Entropy (ApEn), Sample Entropy (SampEn), and Fuzzy Entropy (FuzzyEn), is explored to detect and identify a visitor in a home environment. The research has mainly focused on when another individual visits the main occupier, and it is, therefore, not possible to distinguish between their movement activities. The goal of this research is to assess whether entropy measures can be used to detect and identify the visitor in a home environment. Once the presence of the main occupier is distinguished from others, the existing activity recognition and abnormality detection processes could be applied for the main occupier. The proposed method is tested and validated using two different datasets. The results obtained from the experiments show that the proposed method could be used to detect and identify a visitor in a home environment with a high degree of accuracy based on the data collected from the occupancy sensors.

## 1. Introduction

An environment equipped with an appropriate sensor network is often used to monitor and identify activities of daily living (ADL). Such an environment is referred to as an Intelligent Environment and is used to support independent living. This is often the preferred solution for many older adults who want to live safely and independently in their own home. An automated monitoring system which could also identify abnormalities within the ADL would require an accurate recognition of human activities. Hence, Human Activity Recognition (HAR) has gained increasing attention in recent years [1,2,3]. So far, research related to HAR has devoted particular attention to monitoring and recognizing the human activities of a single occupant in a home environment, in which it is assumed that only one person is present at any given time [4,5,6,7,8,9,10]. However, living environments are commonly inhabited by more than one individual and/or with pet animals [7,10,11].

In a home environment occupied by a single elderly person, it is very likely that there will be visitors/carers who visit the older adult on a regular basis. Therefore, it is essential to identify and distinguish the activities that occur within a multi-occupant environment [12], since multi-occupancy scenarios are far more realistic. Additional challenges exist when dealing with such environments because existing sensors are incapable of distinguishing who has activated them in the absence of any tagging system [13,14,15]. Considering the negative connotations and privacy issues associated with wearable tags/sensors, the wearable sensors are not widely accepted by older adults [6,7,8,9]. It is always a preferred option to use ambient sensors to detect and recognize multi-occupancy in a home environment [8,16]. Due to these challenges associated with the multi-occupancy environments, the research progress is very slow, and many challenges are still unresolved [4].

The current research acknowledges the challenges of multi-occupancy in HAR [5,7,17]. Such challenges are to find suitable models to represent the data association problem (i.e., the identification of the resident) and to find an activity recognition system that captures different interactions among residents [5,7]. Previous studies report that detecting and identifying a visitor in a home environment based on only binary sensors is a primary challenge as binary sensors are not able to provide any information about the personal identity of who triggered the sensor [10,11]. Some previous studies have used wearable sensors to overcome the problem of detecting and identifying multi-occupancy in a home environment [18,19].

To overcome the challenge of detecting and identifying multi-occupancy in a home environment, an unsupervised method is proposed in this paper, based on entropy measures for detecting visitors. The aim of this research is to investigate whether entropy measures including Approximate Entropy (ApEn), Sample Entropy (SampEn), and Fuzzy Entropy (FuzzyEn) can be used to identify multi-occupancy in a home environment. Furthermore, the research investigates the impact of changing the values of an embedded dimension, *m*, and tolerance, *r*, as parameters required to calculate the named entropy measures.

The rest of this paper is structured as follows: In Section 2, studies related to multi-occupancy activity recognition are presented. The proposed method for identifying multi-occupancy in a home environment based on different entropy measures is described in Section 3. The details of entropy measures are presented in Section 4. Data sources for this study are described in Section 5 followed by the experimental results and analysis explained in Section 6. Finally, the conclusions of this paper are drawn in Section 7.

## 2. Related Work

In this section, a brief review of the related work in the context of activity recognition and the challenge of data association in multi-occupancy are presented.

### 2.1. HAR in Multi-Occupancy Environments

HAR is the process of automatically detecting human actions from the data collected from different types of sensors. HAR is relevant to many applications, such as healthcare and assisted living. Many data mining and machine learning algorithms are widely employed for the classification of human activities in intelligent environments [1,20]. In a recent survey [11], the authors provided an overview of the latest investigations on activity recognition in multi-occupancy environments. Their survey includes the existing approaches and current practices used for activity recognition, such as Hidden Markov Model (HMM), Naive Bayes Classifier (NBC), Conditional Random Field (CRF), and Dynamic Bayesian Network (DBN). Moreover, it outlines data association and interactions between occupants as the main challenges in a multi-occupancy environment. Table 1 provides a summary of the related research studies for multi-occupancy environment in the context of the type of sensors used, data association, and the approaches used as well as the results obtained.

Recent advances in identifying activities in multi-occupant environments are presented in [17]. There are many published papers related to pattern recognition that conducted their research to detect HAR in a home environment using a range of different machine learning techniques, including HMM [34,35]. In [5], the Factorial Hidden Markov Model (FHMM) and Nonlinear Bayesian Tracking method are applied and compared for tracking and recognizing human activity. The FHMM is used to model two separate Markov chains corresponding to two users whereas Nonlinear Bayesian Tracking is used to break down the observation area into the number of users. The authors indicated that the Nonlinear Bayesian Tracking method performs better than FHMM (the performance of Bayesian Tracking was 67.9%, while the performance of FHMM was 59.5%). The authors in [4] proposed a new model based on the Markov Modulated Poisson Process (MMPP), an unsupervised method that detects visitors in a smart home environment occupied by an older adult living alone. The ambient sensors are installed in specific locations to cover most of the movement without affecting the routine activities of the occupier. Multiple datasets are used in their research based on the data collected from two apartments. The results of their study show that when MMPP is applied on both datasets, a recall of 78.4% and a precision of 74.9% were achieved for their first dataset, whereas 80.1% recall and 84.2% precision have resulted from their second dataset.

Using embedded sensors in smartphones, including built-in microphones, to recognize multi-occupancy activities is reported in [36]. A recent survey by [37] presents an overview of wearable sensors and bespoke sensors’ usage in activity recognition of multi-occupant environments. The paper highlights the cooperative interaction activities and complex activity recognition in smart homes. The authors of [21] proposed a hybrid approach to recognizing the complex activities of ADL using a smartphone-based sensor. First, different activities such as walking and sitting are extracted by the smartphone accelerometer data, followed by the application of HMM for each person, while the hidden stats are used for the locations of the occupant. Finally, Coupled Hidden Markov Model (CHMM) is constructed to infer the persons’ activities in a multi-occupancy environment. The hidden stats of the CHMM and HMM refer to the activities, whereas the observations of the CHMM and HMM indicate both the location and posture of the individual. The results obtained with five people demonstrated that their proposed method improves the accuracy up to 70%, compared to 30% when only accelerometer data is used. Nevertheless, the cooperative activities, where many residents work together in a cooperative manner such that each person does certain actions of the same activity or together (e.g., two persons moving a table by holding it by the ends), were ignored in this research.

In [22] the authors present an overview of different classification techniques used to recognize human activity based on wearable sensors. They used four supervised classification techniques namely K-Nearest Neighbor (K-NN), Support Vector Machines (SVM), Gaussian Mixture Models (GMM), and Random Forest (RF) as well as three unsupervised classification techniques namely K-Means, GMM, and HMM. These were compared in terms of correct classification rate, recall, precision, and specificity. The results obtained from their study indicate that the K-NN classifier gives the best performance compared to other supervised classification algorithms, whereas the HMM classifier is the model that provides the best result among the unsupervised classification algorithms.

### 2.2. Data Association in Multi-Occupancy Environments

Many of the previous studies on multi-occupancy HAR have used ambient sensors. In this context, some previous studies have focused on the data association in multi-occupancy environments to recognize the residents [24]. For example, in [23], CRF is applied to deal with the problem of data association in multi-resident activity recognition using the dataset gathered by the Centre of Advanced Studies in Adaptive Systems (CASAS) at Washington State University [38]. The results of the study indicate that data association is a fundamental problem in dealing with a multiple-occupant environment. It also mentions that modelling human interaction is a critical issue when modelling activities in a multi-resident environment. Likewise, in [24], the authors have proposed two HMM models to recognize activities in the multi-resident environment. The first model of HMM is used to identify the resident. The second model is used to identify each of the separate activities. The results of these studies show that the performance of the proposed HMM modules is low because sensors are incapable of distinguishing who has activated them in the absence of any tagging system to distinguish individuals in the environment. A study reported in [25] used Incremental Decision Trees (IDT) to attempt to deal with ADL in a multi-occupancy environment. The proposed method is evaluated using the ARAS dataset, a real dataset collected from a smart home environment. However, the results showed that only about 40% rate of classification is achieved.

Most of the previous studies disregarded the possible interactions between occupants due to the data association problem when recognizing multi-occupancy activities [14,26,27,39]. The authors in [14,26] used two different activity recognition models, such as HMM and CRF; whereas the study in [27] used five different classifiers, namely HMM, Decision Trees (DT), KNN, Time-Delay Neural Network (TDNN), and Multi-Layer Perceptron (MLP) to evaluate the activity recognition performance of a single resident in the multi-occupancy environment. They used these methods to recognize multi-occupancy activities by combining labels. The results of this research showed that the TDNN method gives the best performance in terms of accuracy and precision compared with the other methods.

Considering the literature review conducted in this research, there is no reported investigation into the application of entropy measures for identifying ADLs in multi-occupant environments.

## 3. Methodology

It can be argued that the ADLs of a single user in a home environment are different from the ADLs representing multi-users in the same environment. The pattern of activities when a visitor comes to visit an individual (represented as a multi-occupancy environment) will be different from when only the main occupier is in that environment. When the environment is occupied by one person, it is possible to recognize different activities and develop a method representing the normal activities. Once a newly perceived activity differs from the routine for a specific person, that will be represented as an abnormality in the behavior. However, when there is more than one person in the same environment, the activities of the main occupier cannot be easily distinguished from simultaneous activities. The challenge of the research is to identify the activities of the main occupier without introducing any new hardware (or monitoring devices) to the environment or using tagging systems (such as pendant or wristband with RFID). It is possible to use activity count and sensors activation as a measure of multiple occupancy. However, it is not possible to distinguish the visitors. Our earlier work has concluded the maximum sensor activation is not sufficiently reliable enough to distinguish the visitors. It can also be argued that when different type of sensors such as pressure sensor on beds or sofa or door entry sensors are used, then the collected data are not comparable and activity count is meaningless. Therefore, using different measures of changes and/or disorders, including entropy measures, are to be considered since the level of disorder in a multi-occupancy environment is expected to be higher than the single-occupancy case. The proposed approach is based on the hypothesis that the presence of a visitor can be detected when the entropy value is greater than a nominal value. A large value of entropy does not exclusively signify the presence of a visitor in a home environment. For example, a large value of entropy may be influenced by other factors, such as house-cleaning duties, which are different to having a visitor. Having a visitor is considered a deviation in the normal pattern of daily activities for a single person living alone.

A schematic diagram of the proposed framework in this work for identifying a visitor is illustrated in Figure 1. There are three distinct phases to identify the multi-occupancy.
In the first phase, the sensor data representing ADL in a multi-occupancy environment is collected. We are primarily concentrating on the movement data representing the occupancy of different areas in a home environment. Without loss of generality, data gathered from other sensors, including door entry sensors, could also be used. The required numerical features to be used for calculating the sequences of the input vector are extracted from the raw data. The values of this vector are used as inputs to the entropy measures. The selected features representing the ADL from the sensor data are: the start time of entering each location (room), the time spent in each room, and the transitions from one room to another inside the house. The example provided in Section 5.2 will elaborate on the details of these features.In the second phase of the proposed process, different entropy measures are applied to the vector sequences to measure the activity level of multi-occupancy in the home environment.In the third phase, the standard deviation of the entropy measures is used to decide whether there is a visitor to the home environment.

## 4. Entropy Measures

Entropy is a measure that reflects the degree of randomness or uncertainty in a system. It is considered one of the statistical measures to compute the degree of uncertainty in the data [40]. For example, Shannon Entropy defines entropy in terms of a discrete random variable. However, to evaluate the relevance of entropy measures in ADL, different types of entropy measures, namely Approximate Entropy (ApEn), Sample Entropy (SampEn), and Fuzzy Entropy (FuzzyEn), are used to identify a visitor in a home environment. These three entropy measures are more relevant for the binary signals gathered from an Intelligent Environment [41,42]. A brief description of these measures is provided below:

### 4.1. Approximate Entropy (ApEn)

Approximate Entropy was initially introduced by Pincus [43] to classify the concept of complex systems. It is a technique used to quantify the concept of regularity and uncertainty within a data sequence in a system [42]. High regularity and low randomness in the data produce smaller entropy values whereas less regularity gives higher entropy values. The following is the explanation of the procedure for the ApEn-based algorithm as described in [43].

For time series $A=\left[a(i):1\leq i\leq N\right]$ with *N* samples, the sequences of vector $A_{i}^{m}$ can be written as:(1)$$A_{i}^{m}=\left[a(i),a(i+1),\ldots ,a(i+m-1)\right],\;for\;i=1,\ldots ,\left(N-m+1\right)$$
where *m* is the embedding dimension. The distance between two vectors $A_{i}^{m}$ and $A_{j}^{m}$ is represented as the maximum absolute variation between their scalar components:(2)$$d\left[A_{i}^{m},A_{j}^{m}\right]=\max_{k=0,1,\ldots ,m-1}\left(|a(i+k)-a(j+k)|\right)$$

For each $A_{i}^{m}$, the number of j≤N−m+1 such that $d[A_{i}^{m},A_{j}^{m}]\leq r$, where *r* is the tolerance, is given as $N_{i}^{m}\left(r\right)$. The parameters $C_{i}^{m}\left(r\right)$ are then estimated as:(3)$$C_{i}^{m}\left(r\right)=\frac{1}{\left(N-m+1\right)}N_{i}^{m}\left(r\right)$$
where $C_{i}^{m}\left(r\right)$ represent the number of j≤N−m+1 such that $d[A_{i}^{m},A_{j}^{m}]\leq r$. The $\phi^{m}\left(r\right)$ represent the mean value of parameters $C_{i}^{m}\left(r\right)$, which can be defined as:(4)$$\phi^{m}\left(r\right)=\frac{1}{N-m+1}\sum_{i=1}^{N-m+1}\ln C_{i}^{m}\left(r\right)$$

Using $\phi^{m}\left(r\right)$ and $\phi^{m+1}\left(r\right)$, the ApEn (m,r) is defined as:(5)$$ApEn\left(m,r\right)=\lim_{N\to \infty }\left[\phi^{m}\left(r\right)-\phi^{m+1}\left(r\right)\right]$$

Finally, the ApEn is calculated for finite time series length *N* as:(6)$$ApEn\left(m,r,N\right)=\phi^{m}\left(r\right)-\phi^{m+1}\left(r\right)$$

### 4.2. Sample Entropy (SampEn)

Sample entropy was introduced by Richman and Moorman [44]. It is a method to measure regularity and complexity in time series data, which is mostly used for nonlinear analysis. To compute the SampEn, the parameters of embedding dimension *m* and tolerance *r* are required to be defined [45].

For vector sequences, the distance between the two vectors, $d\left[A_{i}^{m},A_{j}^{m}\right]$ are calculated as in ApEn. For a given $A_{i}^{m}$, we calculate $b_{i}^{m}\left(r\right)$ as ${\left(N-m-1\right)}^{-1}$ multiplied by the number of $A_{j}^{m}$ within *r* of $A_{i}^{m}$, where *j* ranges from 1 to N−m and (j≠i). $b^{m}\left(r\right)$ is then calculated as:(7)$$b^{m}\left(r\right)=\frac{1}{N-m}\sum_{i=1}^{N-m}b_{i}^{m}\left(r\right)$$

Similarly, by increasing the embedding dimension *m* to m+1, the $a_{i}^{m}\left(r\right)$ is defined as ${\left(N-m-1\right)}^{-1}$ multiplied by the number of vectors $A_{j}^{m+1}$ within *r* of $A_{i}^{m+1}$, whereas *j* ranges from 1 to N−m and (j≠i). Furthermore, $a^{m}\left(r\right)$ is defined as:(8)$$a^{m}\left(r\right)=\frac{1}{N-m}\sum_{i=1}^{N-m}a_{i}^{m}\left(r\right)$$

Therefore, the probability that two vectors will be matched for *m* samples is given by $b^{m}\left(r\right)$, while $a^{m}\left(r\right)$ represents the probability that two vectors will be matched for [m+1] samples. Then, sample entropy can be calculated as:(9)$$SampEn\left(m,r\right)=\lim_{N\to \infty }\left(-\ln\left[\frac{a^{m}\left(r\right)}{b^{m}\left(r\right)]}\right]\right)$$

SampEn is defined for finite time series length *N* as:(10)$$SampEn\left(m,r,N\right)=\ln\left[\frac{a^{m}\left(r\right)}{b^{m}\left(r\right)]}\right]$$

### 4.3. Fuzzy Entropy (FuzzyEn)

ApEn and SampEn produce matching vectors with either 1 or 0 values. This is not realistic when dealing with real-world examples where the partition between classes may be cryptic or uncertain. Therefore, in the case of SampEn and ApEn, therefore, the input patterns cannot be determined in which the input pattern is placed [42]. To overcome such cases, the fuzzy sets and membership degree are introduced. This membership degree is introduced by a fuzzy membership function μc(x) which allows each point *x* to be associated with a real value within a range [0,1]. The theory introduces a mechanism to measure the degree to which a pattern belongs to a given category, so the membership degree of *x* in dataset *C* will become higher when the value of μc(x) is nearer to unity. Fuzzy entropy was proposed by Chen et al. [46], which is defined as a method to compute regularity in time series. In FuzzyEn, the concept of exponential function exp$\left(-{\left(d_{ij}^{m}\right)}^{n}/r\right)$ is used as a fuzzy function that evaluates the similarity degree of two points (vectors). Such SampEn, FuzzyEn excepts self-matches and beholds only the first (N−m) vectors of length *m* to confirm that $A_{i}^{m}$ and $A_{i}^{m+1}$ are determined for all (1≤i≤N−m). Where $a_{0}\left(i\right)$ is the average value of $A_{i}^{m}$ over the set of *m* values defined as:(11)$$a_{0}\left(i\right)=\frac{\sum_{j=0}^{m-1}a\left(i+j\right)}{m}$$

The distance between vectors $A_{i}^{m}$ and $A_{j}^{m}$ is given by $d_{ij}^{m}$ and calculated as:(12)$$d_{ij}^{m}=d\left[A_{i}^{m},A_{j}^{m}\right]=\max_{\left(k=0,1,\ldots ,m-1\right)}\left|\left(a\left(i+k\right)-a_{0}\left(i\right))-(a\left(j+k\right)-a_{0}\left(j\right)\right)\right|$$

According to the fuzzy membership function $\mu \left(d_{ij}^{m},r\right)$, the similarity degree $D_{ij}^{m}$ between vector $A_{i}^{m}$ and next vector $A_{j}^{m}$ is defined as:(13)$$D_{ij}^{m}=\mu \left(d_{ij}^{m},r\right)$$
where the fuzzy membership function $\mu \left(d_{ij}^{m},r\right)$ is the exponential function defined as:(14)$$\mu \left(d_{ij}^{m},r\right)=exp\left(-{\left(d_{ij}^{m}\right)}^{n}/r\right)$$
where *n* and *r* are the gradient and width of the exponential function, respectively.

For each vector $A_{i}^{m}\left(i=1,\ldots ,N-m+1\right)$, averaging all the similarity degree of its next vectors $A_{j}^{m}\left(j=1,\ldots ,N-m+1,and\;j\neq i\right)$ is defined as:(15)$$\phi_{i}^{m}\left(r\right)=\frac{\sum_{j=1,j\neq i}^{N-m}D_{ij}^{m}}{N-m-1}$$

Then, construct as:(16)$$\phi^{m}\left(r\right)=\frac{\sum_{i=1}^{N-m}\phi_{i}^{m}\left(r\right)}{N-m}$$
and for $A_{i}^{m+1}$, averaging all the similarity degree of its next vectors is defined as:(17)$$\phi^{m+1}\left(r\right)=\frac{\sum_{i=1}^{N-m}\phi_{i}^{m+1}\left(r\right)}{N-m}$$

The FuzzyEn(m,r) is then calculated as:(18)$$FuzzyEn\left(m,r\right)=\lim_{N\to \infty }\left[\ln\phi^{m}\left(r\right)-\ln\phi^{m+1}\left(r\right)\right]$$

Finally, the fuzzy entropy can be defined for finite time series of length *N* as:(19)$$FuzzyEn\left(m,r,N\right)=\ln\phi^{m}\left(r\right)-\ln\phi^{m+1}\left(r\right)$$

## 5. Data Sources and Preprocessing

To evaluate the proposed concept in identifying the ADL in a multi-occupancy environment, two different datasets are investigated. Details of the datasets are presented below followed by the evaluation results in the next section.

### 5.1. Datasets

**Dataset A** representing the ADL is generated using the HOME I/O 3D simulation environment [47]. The virtual house in HOME I/O 3D is equipped with several simulated sensors such as Passive Infra-Red (PIR) motion detectors, door entry sensors, humidity sensors, temperature sensors, and light sensors. These sensors track the occupant’s interaction in different locations in the home. A sample of the simulated environment floor plan and the sensor locations are shown in Figure 2. In our investigation, only the PIR sensors representing the occupancy in an area of the house are used.

To compare the occupancy data for both a single and multi-occupancy environment, firstly, the PIR data representing the room occupancy activities of a single resident for a period of three days is collected. Then, the data for a second person (visitor) is injected into the existing data. The visitor enters the same environment three times during the third day and stays there for a limited time period. The data include date and time of an event when a sensor is triggered, and the location of the sensor. Figure 3 shows a sample of the gathered binary data from the PIR sensors in various locations over one-day period.

**Dataset B** is a dataset publicly shared through the University of California Irvine (UCI) Machine Learning Repository [48]. This dataset includes information related to the ADLs performed by two residents daily in their own homes. This ADL data is collected over 35 days and they are fully annotated. To have a dataset representing multi-occupancy scenarios, a synthetic dataset simulating a visitor is injected into the datasets, which represents a visitor who comes 3 times a week and stays in the house for a couple of hours. The visitor comes around 12 PM and 8 PM. There is variability within the times and the periods of the visits and some days the visitor comes one hour or two hours early or late.

### 5.2. Data Preprocessing

The sensor data representing the activities are logged. Once the sensor data is obtained, the daily behavior features for a resident are calculated, including the start time of entering each location (room), daily pattern, the time spent in each room and the transitions from one room to another inside the house. To make the data suitable for entropy measurement, the dataset is transformed into a set of data points equally separated in time, which is dependent on the computational time of entropy measures. The entropy measures, namely ApEn, SampEn, and FuzzyEn, are used as the measurement of the normal and abnormal patterns of the daily routines. To obtain the entropy value, a vector sequence, called encoded daily activity sequence, is used as an input to the entropy measures. The encoded daily activities sequence is the collection of the activated sensors’ locations in different times, in which each location (room) is given an odd number (e.g., Living room = 1, Bedroom = 3, ...., Other = 9). It was considered that the higher numbers are related to rooms that were frequently used for shorter time periods (here, the corridor). The algorithm implemented in Python provided in [49] was used to measure the normal/abnormal patterns and the degree of difference between the measurements in consecutive days to identify the multi-occupancy patterns.

To clarify the process, a step-by-step example is provided below. Firstly, the daily activity sequence from ADL-labelled sensory data is extracted. The daily activity sequence is encoded by giving each location (room) an odd number as shown in Table 2. The extracted features with the encoded daily activities are used as input vector sequences for the entropy measures. Let us consider N=60 samples (per minute), which is the computational of entropy measure. This period is from 09:00 to 10:00 so the activity sequence vector $A_{N}$, which consists of 60 samples of the encoded daily activity equally spaced in time, is defined as:$$A_{N}=\left[3,3,3,3,3,9,1,1,1,1,9,1,1,1,1,1,1,1,1,\ldots ,1\right]$$

As can be seen from the vector sequence $A_{N}$, the repetition of the same number reflects the time spent in each room (duration). This process will be repeated every hour to calculate the values of entropy measures. Regarding the multiple sensors activated at the same time, only one (i.e., the first activated) sensor at each time will be taken to calculate the vector sequences as shown in Table 2. For example, living room and bathroom are activated at the same time so the first sensor is used in the vector sequence $A_{N}$.

## 6. Experimental Results and Analysis

To evaluate the performance of the entropy measures, two experiments are conducted using two different datasets, as described in Section 5. Furthermore, the robustness analysis is also presented here by applying changes to the embedded dimension *m* and the tolerance *r* in dataset B.

### 6.1. Experiment with Dataset A and Results

The aim of this experiment is to determine whether entropy measures can be used to identify multi-occupancy in a home environment using dataset A. To perform the entropy measures, the dataset is transformed into notional values as described in Section 5.2. The ApEn, SampEn, and FuzzyEn are computed at time intervals for a set of data with different patterns of ADL. The computational period of entropy measures is divided into time slices of lengths 120, 60, and 15 min. The reason for limiting these time slices is that the period of the visits (by the carer) is one hour or less. The values of the parameter *m* and *r* used in Equations (Equation 6), (Equation 10), and (Equation 19), which are needed for entropy calculations, are 2 and 1 respectively. The comparison between these measures is shown in Figure 4.

ApEn, SampEn, and FuzzyEn entropy measures present similar patterns. In addition to entropy measures, the average values of daily pattern and threshold based on the standard deviation of the occupancy data is used in conjunction with the entropy measures for a period of 24 h to decide whether or not there is a visitor in the home environment. For example, the threshold value for this experiment is chosen based on the standard deviation that varies over time and it is not a constant value (it was 0.04). Therefore, when the entropy value of each day goes beyond this value, it means the event is detected as a visitor in the home environment.

The classification performance of ApEn, SampEn, and FuzzyEn measures are evaluated by a confusion matrix that includes accuracy, recall, and precision. There are four possible results for testing the detection of a visitor in the home environment, which are presented as follows:True Positive (TP) is a set of data that contains a visitor event and was correctly classified as a visitor event.False Positive (FP) is a set of data that does not contain a visitor event, but it was incorrectly classified as a visitor event.True Negative (TN) is a set of data that does not contain a visitor event and it was correctly classified as a non-visitor event.False Negative (FN) is a set of data that contains a visitor event and it was incorrectly classified as a non-visitor event.

The accuracy, precision, and recall are computed for each entropy measure. The accuracy is defined as the percentage of correctly identified events (visitor and non-visitor). Precision indicates the percentage of the positive visitor events that are correctly identified, while recall indicates the percentage of true activity labels which were correctly identified. The accuracy of the entropy measures would be high even if the visitor was not well identified. However, the recall and precision would be low. In this case, to show the classification performance of ApEn, SampEn, and FuzzyEn, the precision and recall are chosen as the best choice rather than accuracy to demonstrate the ApEn, SampEn, and FuzzyEn performances.

Table 3 represents the classification performance of ApEn, SampEn, and FuzzyEn using dataset A when calculated at 120, 60, and 15 min time periods. The accuracy, precision, and recall results show that ApEn and FuzzyEn perform much better than SampEn. It is also noted that the best performance is obtained when the computational time is performed based on a one-hour time period. Moreover, the results demonstrate that ApEn and FuzzyEn produce similar results for the accuracy, precision, and recall, which means that the presence of a visitor was accurately identified in the home environment. Therefore, the results of accuracy, precision, and recall indicate that ApEn and FuzzyEn are the best measures for identifying multi-occupancy in a home environment with relatively high accuracy. In contrast, when the calculation period of entropy measures was 15 min, the results show that all three entropy measures achieve a very low performance. This can be justified by the fact that decreasing the calculation period will reduce the number of observations per time period, which will increase the variance. Consequently, the number of false positives will increase, which reduces precision.

### 6.2. Experiment with Dataset B and Results

In this experiment, ApEn, SampEn, and FuzzyEn measures are applied to dataset B. Figure 5 illustrates the results obtained by applying FuzzyEn on dataset B based on one-hour time periods, as well as the FuzzyEn values for each day and the mean value of FuzzyEn for seven days. The mean of FuzzyEn for seven days and threshold based on standard deviation are calculated for a period of 24 h. The threshold value is chosen based on the standard deviation that varies over time and not as a constant value. Therefore, the threshold value for this experiment is chosen based on the standard deviation, which was 0.074 changing over time. The standard deviation is depicted by the dotted line in Figure 5. To detect the visitor in a home environment, FuzzyEn values for each day in Figure 5 were compared with the upper standard deviation boundaries to see which days go beyond the upper boundary of standard deviation. Figure 6 shows the visiting time in each day based on entropy values after they are compared with the upper boundary of standard deviation in Figure 5. As can be seen in Figure 6a–c, there are two bumps in the entropy values, which indicate that the visitor came at those times. This also confirms that the visitor came twice in day 2, 4, and 6. However, Figure 6d shows that no visitor came in day 1, 3, 4, and 7 because there are no bumps in the entropy values. In summary, it can be confirmed that the visitor came twice a day, three days a week, which means that the visitor was identified accurately in all instances.

Table 4 represents the classification performance of ApEn, SampEn, and FuzzyEn using dataset B when they are computed at 120, 60, and 15 min time period. When the calculation period of entropy measures was two hours, the precision results show that the proposed ApEn and FuzzyEn perform much better than SampEn; while the results related to the recall demonstrates that the FuzzyEn performs much better than ApEn and SampEn with a difference of 28% and 14% respectively. Moreover, it is noted that the best performance is obtained when the computational time is performed based on one-hour time periods. The precision results indicate that the ApEn and FuzzyEn are outperformed by SampEn by approximately 38%; whereas the results related to the recall show that ApEn achieves a very low performance compared to FuzzyEn and SampEn by approximately 12.5%. On the other hand, the results show that all the entropy measures achieved a very low performance when they are calculated at 15 min time periods. The results related to precision illustrate that all the entropy measures show a very low performance, which means that the number of false positives increased. To explain what led to these results, we noticed that when the computational time of entropy measures is decreased, the number of observations per time period will be reduced, increasing the calculated variance. Consequently, the number of false positives (detected as visits) increases, reducing the calculated precision.

### 6.3. Robust Analysis

To evaluate the robustness of the proposed method described in this work, the impacts of the parameters *m* and *r* on the classification performance of the entropy measures are investigated. The selection of parameters *m* and *r* needed for the computations of the named entropy measures may be different when they are applied to the ADLs datasets. To investigate the impact of changing the values of these parameters, the performance of the algorithm is examined by using dataset B.

Table 5 shows the results of the experiment in terms of the effect of changing the parameter values *m* and *r* required for the computation of ApEn, SampEn, and FuzzyEn measures using dataset B. Clearly, the result of precision and recall show that the best results obtained when the value of *m* is 2 and *r* ranges from 0.2 to 1.8 respectively. Based on the current results, it appears that when *m* and *r* values are increased, the performance of the algorithm is decreased. To summarize, the algorithm of ApEn, SampEn, and FuzzyEn are affected by the choice of parameter values *m* and *r*.

Based on the results obtained from both experiments, the best performance is obtained when the computational time is performed based on one-hour time periods. Therefore, it is helpful to evaluate how a visitor can be identified when a different shifting of computational time is considered. Table 6 represents the classification performance of ApEn, SampEn, and FuzzyEn using dataset B when they are computed at different shifting time. It is observed that the best performance of ApEn, SampEn, and FuzzyEn is obtained when the computational time is performed based on one-hour time periods with no shifting time and overlapping, x≥30%. The percentage of overlapping, *x*, is calculated as:(20)$$\begin{array}{c} x=\frac{\mathrm{Overlap}\;\mathrm{of}\;\mathrm{the}\;\mathrm{visitor}\;\mathrm{period}}{\mathrm{Calculation}\;\mathrm{period}}\times 100 \end{array}$$

This means that by using the calculation period of one hour without shifting, the visitor can be accurately identified. On the other hand, when the shifting time of 15, 30, and 45 min are used to calculate the entropy measures, the results show that all entropy measures achieved a very low performance with less precision. This can be justified by the fact that when the value of *x* is decreased, the number of false positives will be increased, which reduces the precision. This means that the proposed methods can be used for identifying the visitor if the time period of the visitor per one hour and overlapping is ≥30%.

According to the results shown in Table 3 and Table 4, the best performance is obtained when the computational time is performed based on one-hour time periods. Moreover, the results related to the precision demonstrate that ApEn and FuzzyEn perform much better than SampEn as 100%, 100% and 61.5% respectively. It can be summarized that the FuzzyEn and ApEn are relatively better indices to identify multi-occupancy in a home environment. This also confirms that entropy measures could be used to distinguish occupancy data in the presence of a visitor in a home environment.

### 6.4. Comparison with Existing Modelling Techniques

To evaluate the proposed method described in this work, the performance of the ApEn, SampEn, and FuzzyEn is compared to other methods that achieve the same goal. The dataset B was applied to the SVM model, as well as the Indoor Mobility (IM) measure and the results were compared with the proposed entropy measures. The IM is defined as the frequency of the transition from room to room in a home environment. Readers are referred to [50] for more details about this measure. The features used as input to the SVM are the start time of entering to each location (room), the time spent in each room (duration), the encoded number of each room and the transitions from one room to another inside the house. The final preprocessing step is to divide the data in two subsets, one with about 70% of the instances to training, and another with around the remaining 30% of instances to testing. Moreover, the proposed entropy measures were compared with another study that used MMPP to detects visits in a home environment [4]. Multiple datasets were used in their research based on the data gathered from binary sensors, which were collected by the authors (note that the authors did not use a public dataset). The results presented in Table 7 show the classification performance of ApEn, SampEn, and FuzzyEn compared with the existing SVM, IM, and MMPP in terms of accuracy, precision, and recall. According to the achieved results, entropy measures are considerably better at identifying and detecting visitors in a home environment. Furthermore, it is shown that FuzzyEn outperforms the other methods, which confirms that entropy measures could be used to detect the visitor in a home environment.

## 7. Conclusions

This paper has investigated a means of distinguishing the ADL when the data represents multiple occupants in an environment. The proposed method is based on entropy measures and the ADL data from PIR occupancy sensors. The threshold, based on the standard deviation of the occupancy data in conjunction with several entropy measures, is used to measure whether an activity is identified as an activity in the presence of a visitor. When the entropy values of each day goes beyond the standard deviation, then the event is associated with the presence of a visitor. In this work, two experiments were conducted using two different datasets. The results obtained from both experiments show that the visitor could be identified with a high degree of accuracy based on the data collected from the PIR sensors. The impact of changing the values of embedded dimension *m* and tolerance *r* on the classification performance of the entropy measures was also investigated. The experimental results show that the proposed method obtained a high identification rate of 100% when *m* = 2 and *r* = 1. It should be noted that the values of ApEn, SampEn, and FuzzyEn are affected by the choice of parameter values *m* and *r*. The conclusion for this investigation is that FuzzyEn and ApEn are shown to be the best entropy measures in identifying multi-occupancy in a home environment. This is a preferred alternative solution compared with using wearable sensors or visual cameras with associated privacy concern.

## Figures and Tables

**Figure 1 entropy-21-00416-f001:**
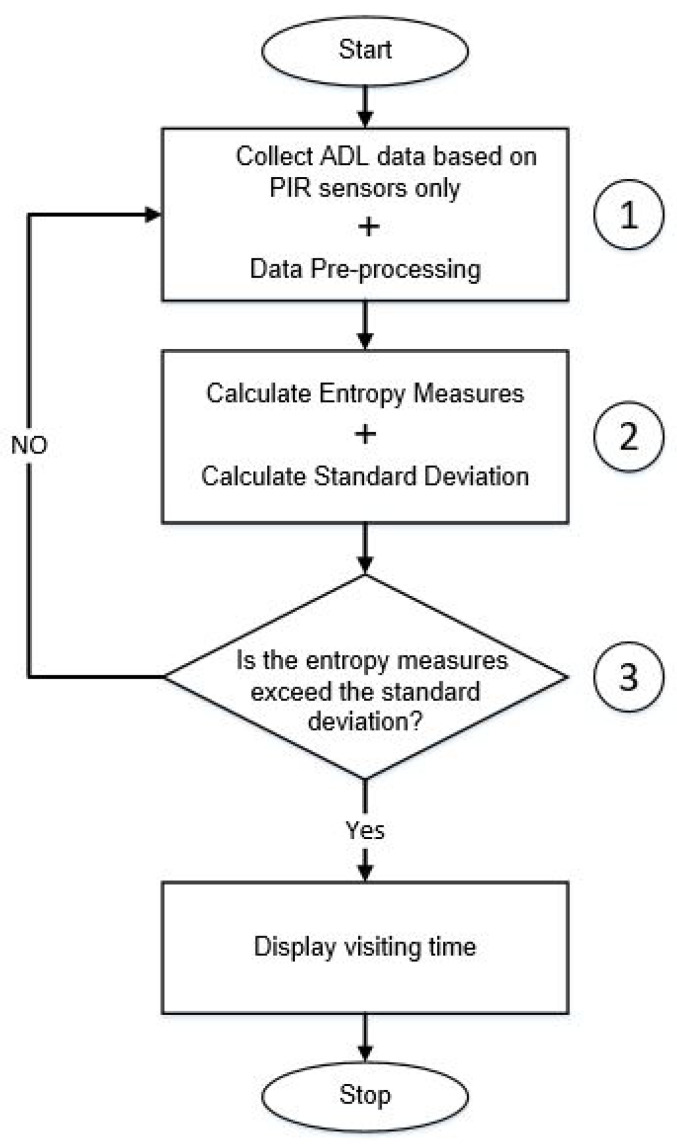
A schematic diagram of the proposed methodology to identify multi-occupancy in activity of daily living.

**Figure 2 entropy-21-00416-f002:**
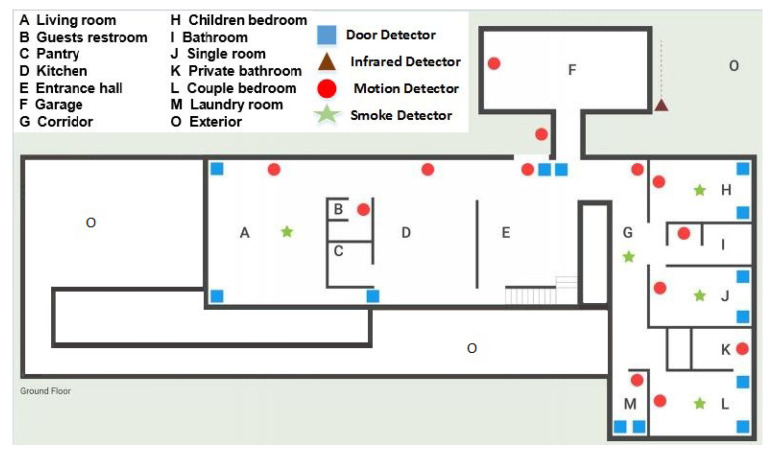
Floor plan and sensors’ location in HOME I/O simulator [47].

**Figure 3 entropy-21-00416-f003:**
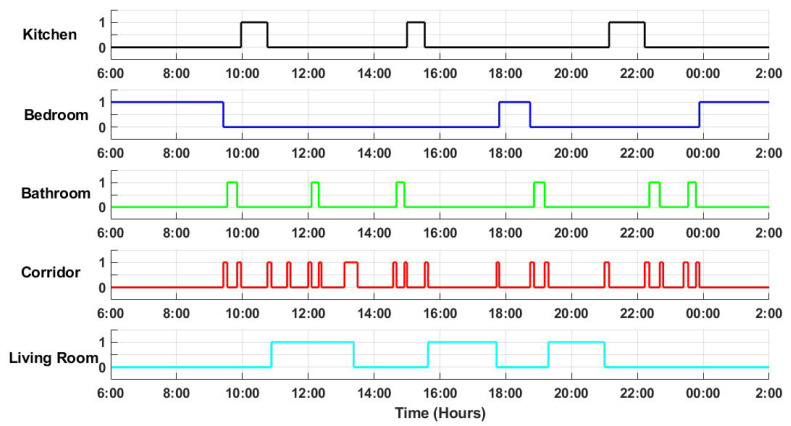
A sample of sensor data gathered from Passive Infra-Red (PIR) sensors in various locations over one-day period, where the y-axis represents the sensor status (on/off) as a binary value in different locations; and x-axis represents time in hours.

**Figure 4 entropy-21-00416-f004:**
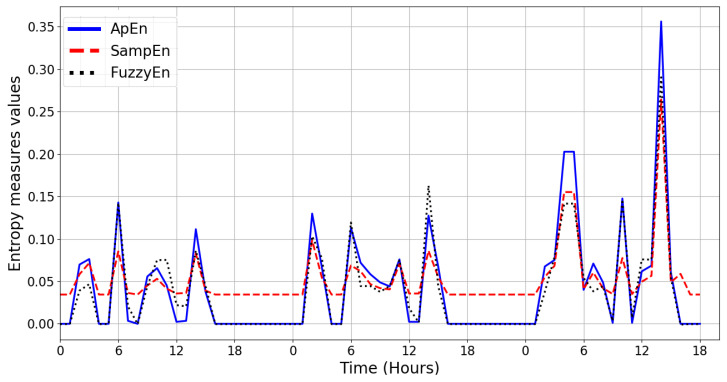
Comparison of Approximate Entropy (ApEn), Sample Entropy (SampEn), and Fuzzy Entropy (FuzzyEn) measures based on the activity of daily living for dataset A.

**Figure 5 entropy-21-00416-f005:**
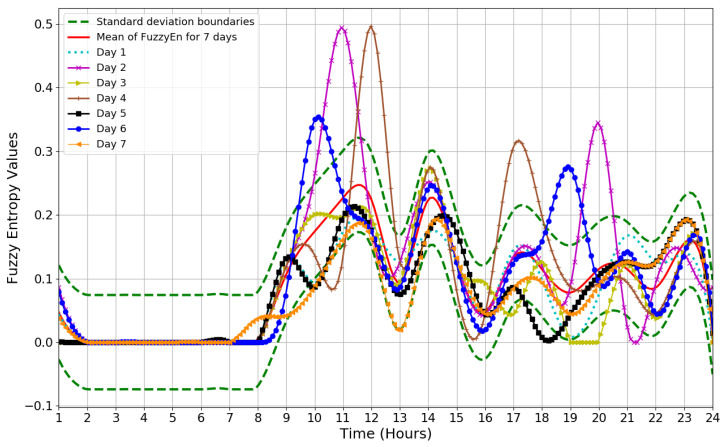
The results obtained by applying fuzzy entropy for seven days based on dataset B. The figure also shows the mean value of fuzzy entropy for seven days and standard deviation boundaries.

**Figure 6 entropy-21-00416-f006:**
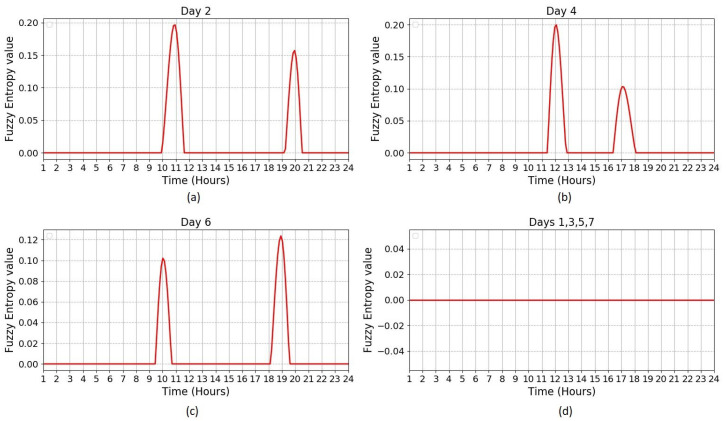
Fuzzy entropy values representing visiting time in each day compared with the standard deviation; (**a**–**c**) show that the visitor came twice in day 2, 4, and 6 because there are two bumps in the fuzzy entropy values; and (**d**) shows that no visitor came in day 1, 3, 5 and 7 because the fuzzy entropy values are zero (no bumps in the entropy values).

**Table 1 entropy-21-00416-t001:** A summary of the related research studies for multi-occupancy environment in the context of the type of sensors used, data association, and the approaches used as well as the results obtained.

Reference	Type of Sensors	Data Association	Approach	Overall Accuracy
[4]	Ambient sensors	yes	Markov Modulated Poisson Process (MMPP)	82.3%
[5]	Ambient sensors	no	Factorial Hidden Markov Model (FHMM) and Nonlinear Bayesian Tracking	64%
[20]	Motion sensor	yes	Finite-set statistics (FISST) and Bayesian filtering	-
[21]	Ambient and smartphone sensor	no	Coupled Hidden Markov Model (CHMM) and HMM	70%
[22]	Wearable sensors	no	K-NN, SVM, GMM and RF	-
[23]	Infrastructure	yes	Conditional Random Field (CRF)	81.3%
[24]	Infrastructure	yes	2 HMMs	84%
[25]	Infrastructure	no	Incremental Decision Trees (IDT)	40%
[14,26]	Infrastructure	yes	HMM and CRF	-
[27]	Infrastructure	yes	HMM, DT, KNN. TDNN and MLP	84.6%
[28]	Motion sensor and wearable	no	Bayesian framework	80.2%
[29]	Video encoder	no	Linear Signal Model for Hybrid and Video Decoding	90%
[30]	Passive sonar	yes	The probabilistic data association (PDA)	85%
[31]	CCTV cameras	no	HMM	98.3%
[32]	Motion sensor	yes	Markov chain Monte Carlo (MCMC)	82%
[33]	Passive sonar	no	SVM	83.5%

**Table 2 entropy-21-00416-t002:** An example of data used to calculate the pre-processed input sequence vector for the entropy measures.

Start Time	Duration (min)	Location	Encoded Number of Each Location
09:00:01	5	Bedroom	3
09:05:22	1	Corridor	9
09:06:00	4	Living room	1
09:06:00	4	Bathroom	5
09:09:59	1	Corridor	9
09:11:00	49	Living room	1
10:00:00	6	Kitchen	7

**Table 3 entropy-21-00416-t003:** The classification performance of Approximate Entropy (ApEn), Sample Entropy (SampEn), and Fuzzy Entropy (FuzzyEn) using dataset A when they are calculated at 120-, 60-, and 15-min time period.

Entropy Measures	Calculation Period of 120 Min			Calculation Period of 60 Min			Calculation Period of 15 Min		
	Accuracy	Precision	Recall	Accuracy	Precision	Recall	Accuracy	Precision	Recall
ApEn	96.5	100	66.6	**100**	**100**	**100**	85.3	35	69.2
SampEn	86.6	50	66.6	96.4	75	75	86	29.3	66.6
FuzzyEn	96.5	100	66.6	**100**	**100**	**100**	87.5	38.2	73.6

**Table 4 entropy-21-00416-t004:** The classification performance of ApEn, SampEn, and FuzzyEn using dataset B when they are calculated at 120-, 60-, and 15-min time period.

Entropy Measures	Calculation Period of 120 Min			Calculation Period of 60 Min			Calculation Period of 15 Min		
	Accuracy	Precision	Recall	Accuracy	Precision	Recall	Accuracy	Precision	Recall
ApEn	96.4	100	57.1	**99.4**	**100**	**87.5**	82.4	8.6	71.4
SampEn	96.4	83.3	71.4	97	61.5	100	87.9	11.4	71.4
FuzzyEn	98.8	100	85.7	**100**	**100**	**100**	85.4	9.6	71.4

**Table 5 entropy-21-00416-t005:** The classification results of the effect of changing the parameter values *m* and *r* required for the computation of ApEn, SampEn, and FuzzyEn measures using dataset B.

m⇒	1		2		3		6		10	
r⇓	Precision (%)	Recall (%)	Precision (%)	Recall (%)	Precision (%)	Recall (%)	Precision (%)	Recall (%)	Precision (%)	Recall (%)
0.2	100	83	**100**	**100**	33.3	83.3	16.6	50	10	50
0.6	100	83	**100**	**100**	33.3	83.3	16.6	50	10	50
1	100	83	**100**	**100**	33.3	83.3	16.16	50	10	50
1.8	100	83	**100**	**100**	33.3	83.3	16.6	50	10	50
2	23.5	66.6	25	66.6	23.5	66.6	14	50	8	33
3	23.5	66.6	25	66.6	23.5	66.6	14	50	8	33
5	14.2	50	11.1	50	10.7	50	7.5	33	6	33

**Table 6 entropy-21-00416-t006:** The classification results of Entropy measures using different shifting time when the computational time is performed based on a one-hour time period.

Results	ApEn	SampEn	FuzzyEn	Shifting Time	Overlapping (*x*) %
Accuracy	99.4	97	100	0 min	x≥30
Precision	100	61.5	100		
Recall	87.5	100	100		
Accuracy	94	95.5	96.4	15 min	16≤x<29
Precision	50	56.2	62.5		
Recall	80	90	100		
Accuracy	88	90.4	86.9	30 min	11≤x<15
Precision	30	35.7	27.2		
Recall	50	41.6	50		
Accuracy	87.5	90.5	91.6	45 min	0≤x<10
Precision	26.2	33.3	39		
Recall	60	60	70		

**Table 7 entropy-21-00416-t007:** Comparison of overall accuracy, precision, and recall for ApEn, SampEn, FuzzyEn, Support Vector Machines, Indoor Mobility and Markov Modulated Poisson Process.

Approach	Accuracy	Precision	Recall
ApEn	99.4%	100%	87.5%
SampEn	97%	61.5%	100%
FuzzyEn	100%	100%	100%
SVM	82.2%	70.8%	72.8%
IM	93.5%	88%	80%
MMPP	78.6	75.2%	78.4%

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
