# Peer review of "Exploring Entropy Measurements to Identify Multi-Occupancy in Activities of Daily Living"

_entropy, 2019, doi:10.3390/e21040416_

Round 1

Reviewer 1 Report

See attached file.

Reviewer 2 Report

Summary

The paper is concerned with an interesting and relevant topic: Identifying phases when multiple persons are present in an environment that is observed by PIR sensors. The authors propose to calculate measures of entropy for segments of the observed sensor data, and classify a segment as "environment occupied by multiple persons", when the entropy exceeds a certain threshold, and "occupied by a single person" otherwise.

Although the approach is interesting and novel (as far as I can tell), the details of the evaluation are incomprehensible in the current form. To me, it is unclear on which type of data the entropy was calculated on (multivariate binary sensor activation, as in Table 1, or on the data as described in Section 5.3); how the threshold value for classification is obtained; why this approach is even necessary (why is it for example not sufficient to just count the number of simultaneously active PIR sensors?).

Also, the results are not compared with any baseline method (like counting the number of simultaneously active PIR sensors; or using a simple HMM), or one of the more elaborate methods described in the related work section. 

Thus, in the current form, the paper is not suitable for publication. Still, I think that the approach might be interesting, and I encourage the authors to submit an improved version. 

Major issues

Goal of the work:

At the beginning of paper, it is unclear what exactly the goal of the paper is: Detecting phases where multiple persons are present in the environment; data association (i.e. maintaining a distribution of identity-track association); multi-target tracking (i.e. tracking the location of multiple objects from noisy measurements)? The goal should be stated more clearly in the abstract and introduction.

Related Work:

The authors do not discuss the rich literature on data association and multi-target tracking. The oldest works go back to the multiple hypotheses tracker (Reid, 1979). Mahler's finite set statistics (Mahler, 2003) is a well-known, systematic extension of Bayesian filtering for multiple targets. Other data association approaches approximate the distributions over permutations by group-theoretic methods (Huang, Guestrin, & Guibas, 2009), use approximations by first-order marginals (Schumitsch, Thrun, Bradski, & Olukotun, 2005), or use methods related to lifted inference in a Bayesian filtering context (Lüdtke, Schröder, Bader, Kersting, & Kirste, 2018).

Additional data association approaches have been proposed by Fortmann, Bar-Shalom, and Scheffe (1983), Han, Xu, Tao, and Gong (2004), and Oh, Russell, and Sastry (2004).

I think it would be beneficial to present a table for the related work: For the mentioned approaches, sometimes accuracy or other performance measures are reported, but reporting them just in the text is quite confusing.

Methodology:

In Section 5.2, it is described that the rooms are encoded into numbers 1 to 9, and the data is transformed into a sequence of those numbers, depending on the currently active PIR sensor. There are multiple issues with this:

What happens when multiple PIR sensors are active at the same time?

Using this approach, a metric is implicitly defined on the rooms, i.e. room 1 and 2 are "more similar" than room 1 and 3. Thus, a movement from room 1 to 2 produces a different entropy value than moving from room 1 to 3. Can you comment on this? If this is the case, I do not see why this would be meaningful in any way. 

It is unclear how the actual classification is performed. In Section 6, it is stated: "So, when the entropy value is higher than a nominal value, it means the event is detected as a visitor in the home environment." What threshold value is chosen? Also, how is this related to the standard deviation that is defined in the same paragraph?

Evaluation:

I am missing a comparison to other methods that achieve the same goal. One straightforward approach would be to use some kind of HMM, where the number of occupants is just another random variable of the system state. Performing inference in this HMM would then directly provide an estimate of the number of occupants. Also, some of the more elaborated methods that are discussed in the related work could be used. 

An even simpler approach is to just look at the number of simultaneously active PIR sensors. If more than one sensor is activated at a given point, more than one person must be present in the environment.

Figure 5 is incomprehensible. What is "Average of Daily profile", "Up StD", "Down StD"? What do the lower four plots show? Also, the plots are labeled "Enteopy".

Other issues

In Section 3, one of the steps in data pre-processing is "extracting the features of daily behavior". I understand that this is explained later on, but at this point, it is completely unclear what happens here (and subsequently, to what kind of data the entropy measures are actually applied). This step should at least be explained briefly here.

In 5.2, the "daily behavior features" are explained in more detail, one of them being "start time of each activity". This is unclear to me. The sensor data consists only of PIR sensor data, so no "activities" are recorded.

Minor issues:

Abstract: The more common term is "human activity recognition", instead of "human activities recognition".

When introducing the entropy measures, it would be good to give some more intuition on the values that are calculated. For example, what does C_i^m(r) and \phi^m(r) represent?

I do not understand this sentence: "In the case of SampEn and ApEn, the input patterns cannot be determined in which input pattern is placed." What exactly is the problem here? Can you maybe give an example?

Round 2

Reviewer 1 Report

See attached file.

Reviewer 2 Report

Summary

The revised submission resolves some of the issues of the previous version. Most notably, the goal of the paper is stated more clearly in the abstract and introduction. However, some issues remain unresolved: It is still not possible to follow the details of the experimental evaluation, e.g. how exactly the data is preprocessed, how the baseline machine learning methods are applied, etc. Furthermore, it seems that very simple methods could be used to solve the problem (like counting the maximum number of simultaneously active PIR sensors), but the authors did not respond to this issue.

Overall, the authors need to work on presenting their results in a much more concrete and precise way. In the current form, the paper is not ready for publication. 

Detailed Remarks

I appreciate that the authors followed the suggestion to present the related work as a table. However, the table does not provide much useful information in the current form. I was expecting the previously present related work to be summarized in a table, not just the papers that were added in this revision. Just writing "Table 1 provides a summary of other various related studies in the context on data association for multi-target tracking" is not sufficient. Furthermore, the column "Approach" is difficult to read, as it is often not clear which lines belong to some approach.

Table 2 and Figure 3 seem to transport the same information, and are therefore redundant. Is Table 2 really necessary, or can it be removed?

For me, it is still not clear how exactly the computation is performed. The authors write: "To obtain the entropy value, the extracted features with the encoding of daily activity sequence are used to calculate the vector sequences that is used as an input to the entropy measures." What is the "encoding of daily activity sequence"? The question regarding what happens when multiple sensors are active simultaneously is not answered in a satisfactory way: Please provide a concrete example of how the preprocessed sequence looks like when multiple sensors are active simultaneously: Is only the sensor returned that was activated most recently? 

The authors state that the standard deviation was used as a threshold for classification, i.e. a constant value for the complete process. However, in Figure 5, the threshold varies over time. Why is this the case?

I still do not understand the plots in Figure 6. In Figure 6, the entropy is almost always close to 0, except when a visitor is present. How is this related to Figure 5, which is supposed to show (as far as I understand) the same entropy data, but there, the entropy looks completely different.

In the current form, the experimental comparison with existing approaches is incomprehensible. It is necessary to explain what exactly the input to the SVM was: Was the preprocessed sensor activation vector used, or the raw dataset, or other features? Which window sizes have been used? 

I am not convinced why it is not possible to use much simpler methods, like taking the maximum number of simultaneous sensor activations.

The authors should carefully check for spelling and grammar errors. Some examples:

"a novel method [...] are explored" -> "a novel method is explored", or "novel methods are explored"

"another individual visit" -> visits

"based on only binary sensors is an essential challenging" -> is challenging

Round 3

Reviewer 2 Report

he revised version of the paper resolves the previous issues.

- Most notably, the explanation regarding the fact that simpler methods could not be used is very helpful. I do not think it is necessary to add the provided figure to the paper. However, I suggest to add a short version of the reasoning (i.e. why counting the number of active sensors cannot be used when multiple sensing modalities are present, etc.) to the main text.

- The other, minor issues have been resolved

Thank you very much for your effort in addressing my concerns. Overall, the paper is ready for publication in the current form (except for the minor point outlined above).
